# Geographic distribution of hospice, homecare, and nursing home facilities and access to end-of-life care among persons living with HIV/AIDS in Appalachia

Sadie P. Hutson[1]*, Ashley Golden[2], Agricola Odoi[3]

1 College of Nursing, University of Tennessee, Knoxville, Tennessee, United States of America, 2 Oak Ridge Associated Universities, Oak Ridge, Tennessee, United States of America, 3 Department of Biomedical and Diagnostic Sciences, College of Veterinary Medicine, University of Tennessee Institute of Agriculture, Knoxville, Tennessee, United States of America

⦿ These authors contributed equally to this work.
* shutson@utk.edu

## Abstract

### Background

Access to healthcare services, from diagnosis through end of life (EOL), is important among persons living with Human Immunodeficiency Syndrome (HIV) and Acquired Immunodeficiency Syndrome (AIDS) (PLWHA). However, little is known about the availability of hospice services in Appalachian areas. Therefore, the objective of this study is to describe the geographic distribution of hospice, homecare and nursing home facilities in order to demonstrate current existence of and access to resources for EOL care among PLWHA in the Appalachian regions of Tennessee and Alabama.

### Methods

This paper reports on the second aim of a larger sequential, mixed methods qualitative-quantitative (qual→quan) study. Data from advance care planning (ACP) surveys were collected by both electronic (n = 28) and paper copies (n = 201) and, among other things, obtained information on zip codes of residence of PLWHA. This enabled assessment of the geographic distribution of residences of PLWHA in relation to the distribution of healthcare services such as hospice and home healthcare services. Hospice and Home Healthcare data were obtained from the Tennessee and Alabama Departments of Health. The street addresses of these facilities were used to geocode and map the geographic distributions of the facilities using Street Map USA. Travel times to Hospice and Home Healthcare facilities were computed and mapped using ArcGIS 10.3.

### Results

We identified a total of 32 hospice and 69 home healthcare facilities in the Tennessee Appalachian region, while the Alabama Appalachian region had a total of 110 hospice and 86 home healthcare facilities. Most care facilities were located in urban centers. The

**Data Availability Statement:** Data has been uploaded here: https://doi.org/10.5061/dryad.t1g1jwt13.

**Funding:** This work was funded by the National Institute of Nursing Research of the National Institutes of Health under Award Number R21NR014055 to SPH. ninr.nih.gov The funders had no role in study design, data collection and analysis, decision to publish, or preparation of the manuscript.

**Competing interests:** The authors have declared that no competing interests exist.

**Abbreviations:** ACP, Advance Care Planning; AIDS, Acquired Immunodeficiency Syndrome; AL, Alabama; CDC, Centers for Disease Control and Prevention; EOL, End-of-life; GIS, Geographic Information System; HIV, Human Immunodeficiency Virus; NAD, North American Datum; PLWHA, Persons living with HIV and AIDS; TN, Tennessee; US, United States.

distribution of care facilities was worse in Tennessee with many counties having no facilities, requiring up to an hour drive time to reach patients. A total of 86% of the PLWHA indicated preference to die at home.

## Conclusions

Persons living with HIV/AIDS in Appalachia face a number of challenges at the end of life that make access to EOL services difficult. Although respondents indicated a preference to die at home, the hospice/homecare infrastructure and resources are overwhelmingly inadequate to meet this need. There is need to improve access to EOL care in the Appalachian regions of both Tennessee and Alabama although the need is greater in Tennessee.

## Introduction

According to the World Health Organization (WHO), access to palliative care is insufficient; globally only 14% of individuals who need care receive it [1]. Presently, there is scant data regarding variations in geographic access to hospice and place of death in the U.S. and globally. In 2012, a study was conducted to explore access to adult inpatient hospice facilities among adults with cancer. The investigators reported that while urban areas of England were well-resourced, large parts of the country had below average accessibility to hospice and above average demand [2]. Chukwusa *et al.* conducted a national population-based observational study in England to explore the urban/rural differences in the relationship between geographic access to inpatient EOL facilities and demonstrated an association between geographic access to inpatient hospice facilities with where people die [3]. This association was stronger for individuals in rural areas where a greater drive time from hospice revealed a higher chance of a home death [3]. In a subsequent study, Chukwusa *et al.* examined regional variations in place of death in North East and South East England, and reported that the further individuals lived from a hospice location, the less likely they were to die in a hospice facility [4]. A recent investigation on the impact of population aging and EOL care in Scotland revealed that by 2040, two-thirds of all deaths could occur in community settings (home or inpatient hospice) [5], demanding an increase in the availability and skills of community-based EOL care facilities and healthcare professionals, respectively. Recent literature that exploring these trends in the U.S. are sparse.

Southern United States (US) has the highest incidence and mortality of Human Immunodeficiency Virus (HIV) infections and stage 3 Acquired Immunodeficiency Syndrome (AIDS) of all causes [6]. With chronic illnesses, such as HIV/AIDS, access to healthcare services is an important part of care from diagnosis, through end-of-life (EOL). Although data from across the US suggests that there is increasing geographic accessibility to hospice in recent years, there is sparse data in the Southeastern Appalachian region regarding access to care for PLWHA. Similarly, little is known about the availability of hospice resources and advance care planning in Appalachian areas. Providing EOL services in rural areas is a challenge due to geography (e.g., long travel times/distances, seasonally poor driving conditions), logistics (e.g., equipment, power, telephone/internet access) and the inability of rural hospices to attract and retain staff. Moreover, stigmatization, isolation and alienation by healthcare, community, and family members related to HIV diagnosis exists [7–10]. Although research shows that accessibility to EOL services and resources may not match rural patients' needs, these studies only examined hospice services [11]. There is a paucity of studies that have explored other EOL

services (e.g., palliative care, transportation, counseling, spirituality) that are essential components of quality EOL care among rural and underserved patients. Moreover, studies investigating patient populations with critical needs for these services such as those with HIV/AIDS in underserved areas such as the Appalachian regions are also lacking.

Increasingly, many large, rural geographic areas of the US lack access to EOL services. However, studies reveal inconsistencies in the number of estimated individuals without access [11, 12]. Virnig et al. showed that most rural areas lacked access to home-based hospice care, and an individual's location was largely associated with hospice access [11]. For instance, 0% of the population within an urban community, 9% of the population in a rural community neighboring an urban community, and 24% of the population in a rural community not adjacent to an urban community lacked access to hospice facilities [11]. In a later study of eight states, 62% to 92% of rural counties had no hospice providers [12]. Looking only at Medicare certified hospice facilities, recent data suggest that 88% of the population lives within thirty minutes of hospice, and 98% of the population lives within an hour of hospice facilities [13].

Assessing geographic accessibility to care facilities is of critical importance for PLWHA in rural Appalachia. While some previous studies investigating geographic accessibility have used straight line (or Euclidean) distances [14, 15] to estimates travel distances, these methods are not as good as methods that estimate travel time. Other studies have used a number of different approaches for estimating travel time to care for other health outcomes ranging from trauma [16], to cancers [17], and emergency care [18–20]. A study by Tansley *et al* in Nova Scotia computed predicted travel times to trauma centers using cost distance analysis [16]. In separate studies Pedigo and Odoi [18] and Busingye, Pedigo and Odoi [20] used network analysis to compute travel times for heart attack and stroke patients in East and Middle Tennessee, respectively. Schuurman *et al* also used network analysis to identify hospital catchment areas in rural areas of British Columbia to describe access to hospital-based care services [21]. Use of travel time estimated from road networks is a better method for assessing accessibility because: (i) it accounts for speed limits [18]; (ii) people relate more easily to travel time than distance when making travel decisions to care; (iii) travel time is a more sensitive measure than distance [22].

Only an estimated 78% of Tennessee's population is within 30 minutes and 98% of the population is within 60 minutes from a hospice. In the Southern region, Tennessee is surpassed only by Kentucky, West Virginia, and Arkansas on both measures for lack of hospice facilities [13]. Despite the fact that the majority of individuals have expressed a desire to die at home and rural individuals expressed a desire for a peaceful death without suffering, hospice enrollment remains low: approximately 6% for adult PLWHA and <1% for pediatric and adolescent patients with AIDS [23–26]. End-of-life services, like hospice, are likely inaccessible within Appalachian regions due to known barriers of rural geographic landscape, remote access of the region and marginalization [9, 10].

In a study evaluating hospice and geographical relationship to cancer decedents, 76% of Alabama counties contained at least one hospice facility with an average distance to hospice care of 9.8 miles. Of note, 52% of patients dying of cancer received hospice care, and of non-users, 77% were within 20 miles of a hospice [27]. The number of individuals with cancer using hospice is significantly greater than PLWHA [25]. While this might suggest that the Southern region is well served with hospice facilities, Tennessee (TN) has more geographic barriers than Alabama (AL) and may, therefore, have more problems with access than AL. Lindley and Edwards found that pediatric hospice accessibility in the eastern region of TN, including Knoxville and areas of Appalachia, remains low with a decreasing number of facilities, despite increasing needs. From 2009 to 2011, the number of hospice providers for

pediatric patients dropped from 42% to 32% while counties needing hospice but lacking the resources increased from five to ten [28]. The observed pattern is the need and growing resources of hospice facilities (approximately 41% from 2000) in urban areas with concurrent increasing need and dissolving hospice resources in rural regions [29].

Due to the increasing need for hospice services and lack of availability, it is imperative that all EOL services be investigated. In particular, PLWHA face barriers beyond geographic access; given that only access to hospice services has been investigated in this population, the objective of this study was to identify and describe the geographic distribution of hospice, homecare, and nursing home facilities in order to assess and identify disparities in and access to a more diverse set of resources for EOL care among PLWHA in the Appalachian regions of TN and AL.

## Materials and methods

### Design

In this paper, we are reporting on the second aim of a larger sequential, mixed methods qualitative-quantitative (qual→quan) study exploring perceptions of EOL needs from the viewpoints of PLWHA, including diverse subgroups living in Appalachian counties in TN and AL. The aims of the parent study were to: I) assess the EOL care and service needs of PLWHA in Appalachian TN and AL, and II) examine geographic access to EOL services for PLWHA in Appalachian TN and AL. For Aim I, we qualitatively interviewed PLWHA to assess EOL care and service needs in the context of physical, psychological, social, spiritual/religious, cultural, and ethical/legal domains. We then used the qualitative data to adapt, validate, and pilot a quantitative advance care planning (ACP) survey in the study region. Simultaneously, we used Geographic Information systems (GIS) to ascertain what the existing EOL resources are in the study areas to address Aim II. Here we report the findings, which reflect the actual EOL care resources available to PLWHA in these remote areas.

### Data collection

Subjects were recruited in partnership with HIV Centers of Excellence, AIDS Service Organizations, support groups, faith-based advocacy agencies, regional health offices, and other advocacy groups in the states of TN and AL. Inclusion criteria were as follows: 21 years of age or older; ability to read speak, and write in English; self-acknowledged diagnosis of HIV infection; and residing in an Appalachian county in TN or AL. The study was approved via a fullboard mechanism of the Institutional Review Boards (IRB) at East Tennessee State University; the University of Tennessee, Knoxville; and the University of Alabama, Tuscaloosa. A questionnaire survey, used to collect quantitative data, was pre-tested in Tennessee, among health professionals involved in the care of PLWHA as well as members of the intended study population of PLWHA. The survey data collection was then conducted between January 2015 and July 2015 using a web-based means via the platform Qualtrics (n = 28) as well as paper-based copies of the instrument (n = 201), based on patient preference and availability/access to the World Wide Web in some of the rural counties. In some cases, community partners distributed the link to the survey to patient groups and/or support groups. In other cases, the study team made themselves available in clinic waiting rooms and invited potential subjects to complete the survey (on an electronic tablet or on paper) as they waited for their appointments. Those who accepted to complete the survey were taken to a quiet, private room from where they completed the survey. Upon request and to facilitate assistance with reading, questions on the survey and answer choices were read aloud by any subject who needed it.

## Data sources

**Advance Care Planning (ACP) survey.** The ACP survey was used to ascertain the geographic distribution of residence PLWHA in relation to clinics. Subjects (n = 229) were asked to provide their zip code for their place of residence. Due to the sensitive nature of the diagnosis and the small number of cases of HIV in some counties, we were unable to ascertain street level data for the purpose of mapping as that would have resulted in identification of study subjects.

**Hospice and home healthcare data.** Hospice and Home Healthcare data were obtained from the Tennessee and Alabama Departments of Health. The facility name, type, and street address data from these facilities were used to geocode and map the geographical distributions of the facilities across the two states under study (Appalachian regions of Alabama and Tennessee). All mapping were done in ArcGIS 10.3 [30].

**Street network data.** The street network data, used for network analysis, were obtained from Street Map USA. These data were critical for network analysis as they provided information on road connectivity, speed limits and other restrictions such as one-way streets and no U-turns. The North American Datum (NAD) 1983 State Plane coordinate system was used. Travel times were computed from the distance and speed limit of each street segment using the field geometry calculator of ArcGIS 10.3 [30]. All cartographic boundary files used to provide base maps for visualizations were obtained from the US Census Bureau website [31]. These files included: (a) county level base maps that were used to overlay street network data and the geographic distributions of healthcare facilities; (b) zip code base map for the network analysis map of survey participants' access to clinics where they indicated they received routine care.

**HIV data.** Aggregated county level HIV prevalence data were obtained from the National HIV Surveillance System, accessed through the Center for Disease Control and Prevention's (CDC) Health Indicators Warehouse [32]. The 2012 rates were the most recently available rates and were expressed as the number of prevalent cases in each county per 100,000 population.

**Geographical information system analysis.** Network analysis was performed, using the service area solver of the network analyst extension in ArcGIS 10.3 [30] to compute driving times from different neighborhoods to hospice and home healthcare facilities in the two states under study. Using Dijkstra's algorithm, the shortest travel paths and minimum travel times from different neighborhoods to the care facilities (hospice and home healthcare) were computed. The travel time limits were set to 15, 30, 45 and 60 minutes. A maximum travel time of 60 minutes was used because this was deemed the longest reasonable time that individuals would travel for regular care [e.g., including nurses driving to patient homes for hospice or homecare visits and family driving to see patients in nursing homes]. The network models used for these computations were specified as follows: (1) U-turns were not allowed to avoid dead-end streets and doubling back on to the same street; (2) distance (in feet) was used as the travel cost (or impedance). This ensured that shortest travel time was taken. Based on these model specifications, 15, 30, 45 and 60- minute travel time buffers were generated to represent travel time zones to the different hospice and home healthcare facilities.

Maps were generated to display the following: (1) geographic distribution of hospice and home healthcare facilities overlaid with travel (driving) times to identify areas that have access to care within 15, 30, 45 and 60 minutes of driving; (2) over lay of the distribution of HIV patients who participated in the study, the location of the clinics they attended and the geographic distribution of drive time identifying zip codes of patients that had access to their clinics within 15, 30, 45 and 60 minutes of driving; and (3) over lay of the county level prevalence

of HIV and all healthcare facilities to investigate if there are any disparities in access to care in relation to prevalence of HIV.

## Ethical approval

The study was approved via a full-board mechanism of the Institutional Review Boards (IRB) at East Tennessee State University; the University of Tennessee, Knoxville; and the University of Alabama, Tuscaloosa. A waiver of informed consent was approved related to the data collected for this portion of the project. Data were analyzed anonymously. UTK IRB-14-08918 B-FB

## Results

Overall, 229 subjects who met inclusion criteria completed the ACP survey across the Appalachian counties of TN and AL (Table 1). Of the subjects completing the survey, eight (8) individuals provided a gender other than male/female (e.g. intersex [n = 1], transgender [n = 2], female-to-male transgender [n = 1], male-to-female transgender [n = 1], other [n = 2], and genderqueer/androgynous [n = 1]); data for these individuals are not included in Table 1. Given that we were unable to ascertain the precise reach of our survey, it is not possible to

**Table 1. Demographic characteristics of survey respondents from Alabama and Tennessee.**

| Factor | Females | | Males | | Total | |
|---|---|---|---|---|---|---|
| | n | % | n | % | n | % |
| **State** | | | | | | |
| Alabama | 38 | 69 | 97 | 58 | 135 | 61 |
| Tennessee | 17 | 31 | 69 | 42 | 86 | 39 |
| Total | 55 | 100 | 166 | 100 | 221 | 100 |
| **Ethnicity** | | | | | | |
| Hispanic or Latino | 0 | 0 | 1 | 1 | 1 | 1 |
| Non-Hispanic or Non-Latino | 55 | 100 | 106 | 64 | 161 | 73 |
| Unknown/Not Reported | 0 | 0 | 59 | 36 | 59 | 27 |
| Total | 55 | 100 | 166 | 100 | 221 | 100 |
| **Race** | | | | | | |
| American Indian/Alaskan Native | 0 | 0 | 0 | 0 | 0 | 0 |
| Asian | 1 | 1 | 1 | 1 | 2 | 1 |
| Native Hawaiian or Other Pacific Islander | 0 | 0 | 0 | 0 | 0 | 1 |
| Black or African American | 34 | 62 | 120 | 72 | 154 | 70 |
| White | 18 | 33 | 35 | 21 | 53 | 24 |
| Other | 2 | 4 | 10 | 6 | 12 | 5 |
| Unknown/Not Reported | 0 | 0 | 0 | 0 | 0 | 0 |
| Total | 55 | 100 | 166 | 100 | 221 | 100 |
| **Education** | | | | | | |
| Less than High School | 17 | 31 | 19 | 11 | 36 | 16 |
| High School Graduate | 22 | 40 | 49 | 30 | 71 | 32 |
| Some College | 9 | 16 | 55 | 33 | 64 | 29 |
| College Graduate | 6 | 11 | 36 | 22 | 42 | 19 |
| Post-grad/Professional | 1 | 2 | 7 | 4 | 8 | 4 |
| Total | 55 | 100 | 166 | 100 | 221 | 100 |

*Not all subjects reported ethnicity.

report an accurate response rate. As a point of context and significance for the results of the GIS analysis below, the ACP survey revealed important findings pertaining to subjects' preferences at EOL. First, 86% of subjects (n = 196) stated they wished to die in their own homes. The majority of the respondents also indicated a desire for healthcare providers who would be able and willing to come to their homes to provide care. To remain at home when death is imminent varies with progression of HIV as well and the presence of comorbidities (52% of respondents [n = 119] reported other health conditions in addition to HIV) might require professional in-home health care, or at least an informal care provider. As many as 44% of the respondents anticipated that there would be nobody to take care of them at EOL (n = 102).

## Geographical distribution of hospice and home healthcare facilities

There were 32 hospice and 69 home healthcare facilities in the Appalachian region of Tennessee while the Appalachian region of Alabama had a total of 110 hospice and 86 home healthcare facilities. As expected, most of the care facilities were located in urban centers. Notably, the disparities in distribution of the care facilities are more dramatic in Tennessee than Alabama. This was evidenced by the fact that sixteen counties in Tennessee did not have either a hospice or a home health care facility. This was not the case in Alabama where only two counties did not have either a hospice or home healthcare facilities (Figs 1 and 2).

## Travel times to hospice and home healthcare facilities

Figs 1 and 2 depict variations in geographic access to home health and hospice care in Tennessee and Alabama, respectively, where darker shades of grey indicate longer travel times. As would be expected, most of the communities that had access to hospice or home healthcare within a 15-minute drive were located in and around urban centers. Although several communities in the Appalachian region of Tennessee had access to either a hospice or a homecare

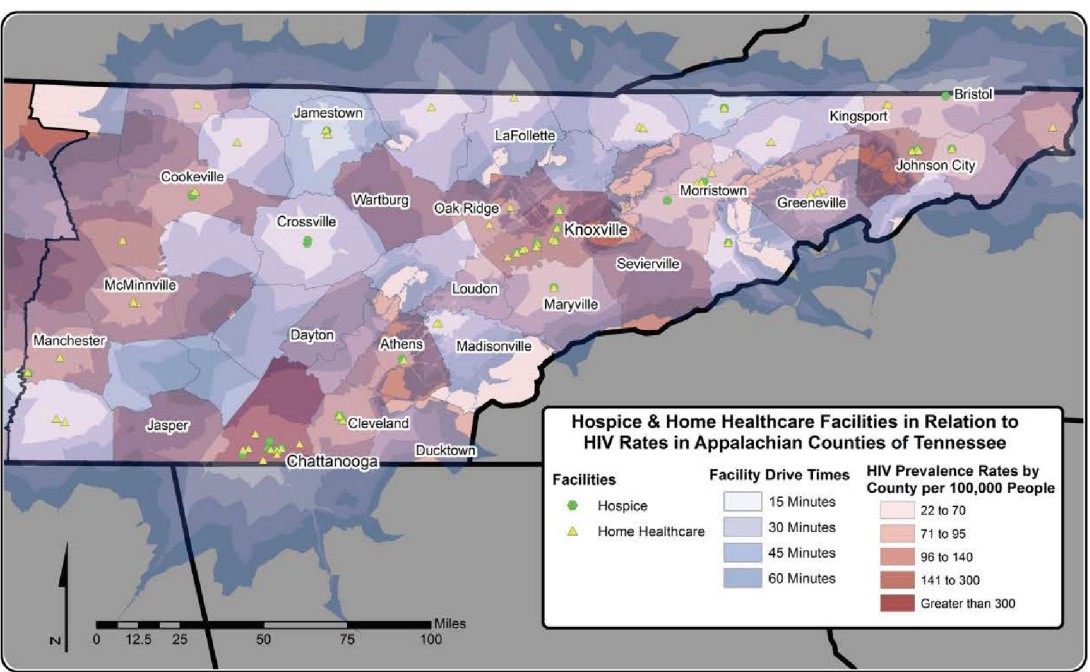

**Fig 1. Travel times to hospice and home healthcare in Tennessee.** "The Figure was created by the authors and published under a CC BY license, with permission from ESRI, original copyright [original copyright year]".

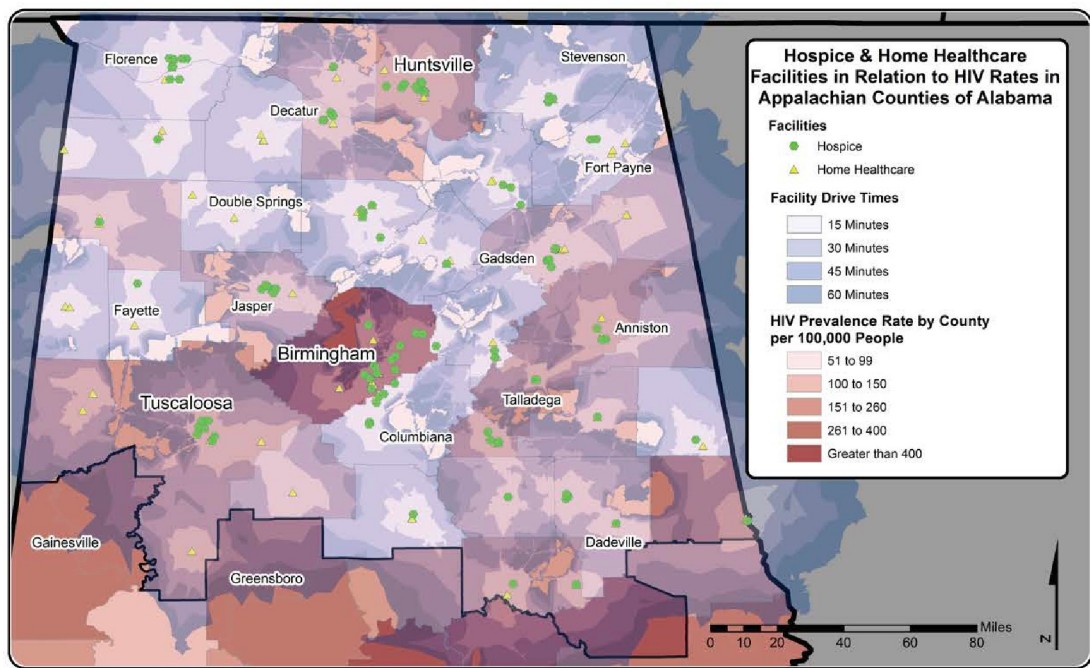

**Fig 2. Travel times to hospice and home healthcare in Alabama.**

facility within a 45-minute drive, several communities had driving times as long 1-hour. This was especially true toward the middle and western parts of Appalachian Tennessee where ten of contiguous counties did not have hospice or home healthcare. This means that patients in the aforementioned area or home healthcare providers would need to cross more than one county border to accomplish access, and provide hospice, or even home health care to these patients (Fig 1).

Although the Appalachian region in Alabama also had similar disparities in drive times with drive times being shorter in urban centers and environs, the disparities were not as extensive as those in Appalachian region of Tennessee (Figs 1 and 2). This is mainly due to that fact that all but two of the Appalachian counties had at least 1 hospice or home healthcare facility, unlike the situation in Appalachian region of Tennessee, where numerous counties did not have hospice or home healthcare. This situation was exacerbated in Appalachian Tennessee in situations where under-served counties were contiguous to each other making for quite long drive times requiring crossing more than one county border to access care.

## Geographical distribution of HIV prevalence estimates in relation to drive times to hospice and home health care facilities

The 2012 HIV prevalence estimates are depicted for each county in Tennessee (Fig 1) and Alabama (Fig 2), where darker shades of red indicate higher prevalence estimates. The prevalence estimates of five counties in TN (Hancock, Grundy, Fentress, Pickett, and Sequatchie) were suppressed due to low numbers of HIV cases (< 5 cases). The prevalence estimates ranged from 22 to 831 per 100,000 (median 93; interquartile range 71–134) in TN and from 53 to 782 per 100,000 (median 170; interquartile range 105–296) in AL. In both TN and AL, the highest prevalence of HIV was observed around the more urban areas of Knoxville, Chattanooga, Johnson City, Birmingham, Tuscaloosa, and Huntsville. However, there were areas of

moderately high prevalence estimates of HIV where access to hospice and home health facilities exceeded 30 minutes, and in some instances exceeded 1-hour drive times. In Tennessee, these areas included: counties in the western part of the Appalachian study area around Cookeville, McMinnville, and Manchester; mountainous areas near Bristol, Johnson City, Greeneville, and Sevierville; and areas where drive time exceeded 60 minutes near Athens and Wartburg. In Alabama, moderately high prevalence estimates of HIV were observed in areas with longer drive times to accessible care in areas surrounding Tuscaloosa, Talladega, Dadeville, and Anniston. High prevalence estimates were observed near Gadsden, Decatur, and Jasper; however, these areas have drive times less than 30 minutes to hospice and home health care.

## Geographic distribution of residence of HIV patients in relation to clinics

**Tennessee.** The geographical distribution of the communities of residence of PLWHA living in Appalachian region of Tennessee who participated in the study, relative to travel time to clinics where they received care, is shown in Fig 3. The majority of the patients tended to live in the more urbanized areas. Although it is clear that the majority of the patients have access to the clinics from where they received care within 30 minutes of travel, this was not always the case. The areas around Athens especially stand out as having relatively high numbers of HIV patients that participated in the study and that had to travel over 1 hour to the clinic where they received care. Other areas where patients had to travel 45 minutes or more for care were Kingsport, Morristown, Cleveland and Jasper (Fig 3).

**Alabama.** Contrary to the findings in Tennessee where most PLWHA participating in the study resided in the urban areas, the majority of PLWHA in Alabama who participated in the study mainly lived in the more rural areas (Fig 4). Due to this rural-urban shift in the

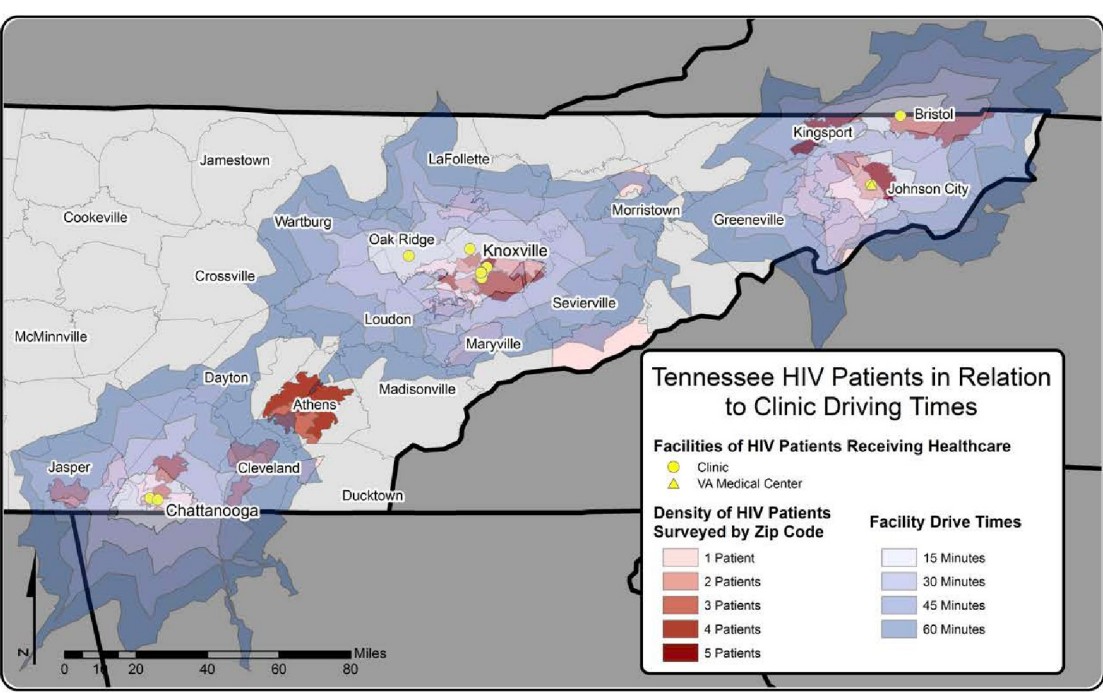

**Fig 3. Geographical distribution of zip codes of residence of HIV study participants in relation to traveling time to the clinics they attended in the Appalachian region of Tennessee.** Note: Data on density of patients in this map is based on respondents to the questionnaire survey.

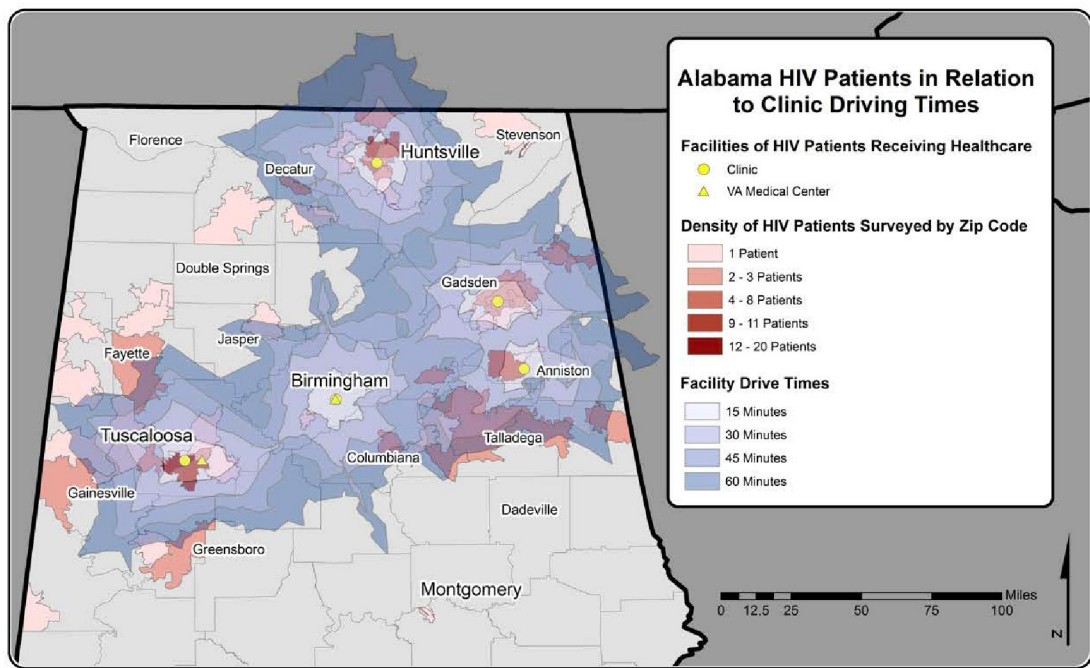

**Fig 4. Geographical distribution of zip codes of residence of HIV study participants in relation to traveling time to the clinics they attended in the Appalachian region of Alabama.** Note: Data on density of patients in this map is based on respondents to the questionnaire survey.

distribution of patients, several PLWHA in the Appalachian region of Alabama had to travel more than 1 hour to the clinic where they received care. Moreover, even among those who had access to their clinics within less than one hour of travel, a significant proportion still had to travel for more than 30 minutes (Fig 4). Worth noting are PLWHA living in Stevenson and most of Greensboro and Fayette who could not reach the clinic they attended within 1 hour of travel. Moreover, PLWHA living in areas such as Jasper, Gainesville, Talladega, Columbiana and Decatur generally had to travel for over 45 minutes to receive care (Fig 4).

## Discussion

The high prevalence of HIV in the southern portion of the US is the basis for the understanding that health care services are vastly needed in the region [6]. Yet, access to services, including EOL care had yet to be investigated in this region. Understanding how Appalachians perceive their health is a dynamic that precedes many other health concerns for this population. 74% of Appalachians consider themselves as healthy, despite opposing medical reports; this raises a concern for the interplay in perceived versus actual health [33]. Noting this discrepancy exemplifies the lack of health education in the region, which could be from a lack of access to services. Similarly, this study demonstrates that access to services for EOL care among PLWHA in the Appalachian region was significantly lower than the level of need. Long drive times to access care services could be over an hour. The long drive time is important to note due to restrictions in rural areas where individuals living with a chronic non-cancer diagnosis, such as PLWHA, have reduced access to specialist palliative care programs [34]. This is in direct contradiction to the services that were preferred, as demonstrated by the survey portion of our mixed methods study. As many as 86% of the respondents indicated wanting to die at home, yet, the majority lived alone; and 44% of them reported that they were concerned that

there would be nobody to care for them at EOL. In order to provide in-home EOL care to PLWHA, they must have access to EOL care resources, including palliative care.

Additionally, the distribution of healthcare services as a whole in the Appalachian region is low. For example, the supply of primary care physicians per 100,000 people is 12% lower in Appalachia than the national average [35]. Not only are primary care services inadequate to meet patient needs, the same is true for hospice, homecare and nursing home services for EOL care, as validated by the use of GIS analysis in this study. The need for EOL services is evident given the moderately high rates of HIV in many areas among individuals who had drive times that exceeded 30–60 minutes to access hospice and home health services. Urban centers were found to house the majority of services. There was a total of 32 hospice and 69 home healthcare facilities in the Appalachian region of TN while the Appalachian region of AL had a total of 110 hospice and 86 home healthcare facilities. Several counties in Tennessee did not have either a hospice or a home health care facility. This was not the case in Alabama where only two counties did not have either a hospice or home healthcare facilities.

Future research is needed to address how access to EOL services can be improved for PLWHA in addition to determining planned expansion for EOL and advance care planning services. Although it is clear that there is a lack of facilities to provide these services in proportion to the number of people who require it, expansion of EOL care across other domains could be an option. Investigating how PLWHA can receive EOL services despite a lack of access to care and someone to care for them is another area that is ripe for future investigations. Further, future studies should explore individuals' understanding of and importance of advance care planning and EOL care among PLWHA as it relates to quality of life.

A major gap exists in the science pertaining to the use of home care facilities on provision of EOL care. Further, while it is important to use large data sets to generate associations between access to EOL care and place of death, this approach does not account for patient preferences about where they wish to die. As such, the current state-of-the-science is inadequate to explain the influence of patient preferences on relationship between place of death and access to EOL care. Considering it is estimated that the aging population will result in an increase in the annual death rate from 38% in 2014 to 53.6% in 2040, EOL care provision must double [36]. The findings from this study demonstrate the critical need for policy development, mandating the integration of hospice services into the financing of healthcare systems at all levels of care, both community-based and in acute care settings.

## Conclusions

We have shown evidence of the disparities between need and access in the Appalachian South regarding hospice, homecare, and nursing home services among PLWHA. One of the most important findings of this study is the fact that although PLWHA in South Central Appalachia prefer to die at home, the hospice/homecare infrastructure and resources are overwhelmingly inadequate to meet these needs. The same geographic situation is faced by thousands of chronically ill persons in Southern Appalachia who may also prefer to die at home. In that sense, these findings are generalizable to a much larger population of individuals. This study was comprehensive in painting a detailed multi-perspective picture of needs, resources, and barriers to care access, based on multiple data sources and mixed methods.

## Acknowledgments

The authors thank the research participants who completed the surveys and the study team for their assistance with data collection.

## Author Contributions

**Conceptualization:** Sadie P. Hutson.

**Data curation:** Sadie P. Hutson.

**Formal analysis:** Sadie P. Hutson, Ashley Golden, Agricola Odoi.

**Funding acquisition:** Sadie P. Hutson.

**Investigation:** Sadie P. Hutson.

**Methodology:** Sadie P. Hutson.

**Project administration:** Sadie P. Hutson.

**Resources:** Sadie P. Hutson.

**Software:** Sadie P. Hutson.

**Supervision:** Sadie P. Hutson, Agricola Odoi.

**Writing – original draft:** Sadie P. Hutson.

**Writing – review & editing:** Sadie P. Hutson, Ashley Golden, Agricola Odoi.

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
