## [Decision Letter · Decision Letter 0]

4 Aug 2020

PONE-D-20-04475

Geographic distribution of hospice, homecare, and nursing home facilities and access to end-of-life care among persons living with HIV/AIDS in Appalachia

PLOS ONE

Dear Dr. Hutson,

Thank you for submitting your manuscript to PLOS ONE. After careful consideration, we feel that it has merit but does not fully meet PLOS ONE’s publication criteria as it currently stands. Therefore, we invite you to submit a revised version of the manuscript that addresses the points raised during the review process.

We look forward to receiving your revised manuscript.

Kind regards,

Tim Luckett

Academic Editor

PLOS ONE

2. Please include in your Methods section the date ranges over which you recruited participants to this study.

3. Please include additional information regarding the survey or questionnaire used in the study and ensure that you have provided sufficient details that others could replicate the analyses. If you developed and/or translated a questionnaire as part of this study and it is not under a copyright license more restrictive than Creative Commons Attribution (CC-BY), please include a copy, in both the original language and English, as Supporting Information.

4.We note that you have indicated that data from this study are available upon request. PLOS only allows data to be available upon request if there are legal or ethical restrictions on sharing data publicly. For more information on unacceptable data access restrictions, please see http://journals.plos.org/plosone/s/data-availability#loc-unacceptable-data-access-restrictions.

6. Please include your tables as part of your main manuscript and remove the individual files. Please note that supplementary tables (should remain/ be uploaded) as separate "supporting information" files.

7.We note that [Figure(s) 1, 2, 3 and 4] in your submission contain [map/satellite] images which may be copyrighted. All PLOS content is published under the Creative Commons Attribution License (CC BY 4.0), which means that the manuscript, images, and Supporting Information files will be freely available online, and any third party is permitted to access, download, copy, distribute, and use these materials in any way, even commercially, with proper attribution. For these reasons, we cannot publish previously copyrighted maps or satellite images created using proprietary data, such as Google software (Google Maps, Street View, and Earth). For more information, see our copyright guidelines: http://journals.plos.org/plosone/s/licenses-and-copyright.

1.    You may seek permission from the original copyright holder of Figure(s) [1, 2, 3 and 4] to publish the content specifically under the CC BY 4.0 license. 

Reviewers' comments:

Reviewer's Responses to Questions

**Comments to the Author**

1. Is the manuscript technically sound, and do the data support the conclusions?

Reviewer #1: Yes

Reviewer #2: Yes

2. Has the statistical analysis been performed appropriately and rigorously? 

Reviewer #1: I Don't Know

Reviewer #2: Yes

3. Have the authors made all data underlying the findings in their manuscript fully available?

Reviewer #1: Yes

Reviewer #2: No

4. Is the manuscript presented in an intelligible fashion and written in standard English?

Reviewer #1: Yes

Reviewer #2: Yes

5. Review Comments to the Author

Reviewer #1: This is an interesting paper and highlights some startling facts about the prevalence of HIV/Aids in this geographical region and the provision available for EOL care.

The data presented are detailed, but difficult for an international audience to easily grasp. The problems addressed are very real. But I think a more detailed theoretical and literature based treatment of the data would be helpful.

I think this is an important public health issue, but wonder of PLOS One is the right place for this to appear. I would hope that you might be able to draw the attention of local public and private service planners in your area. I realise that the group you are looking at is unlikely to be able to access health insurance and so their situation is of even more importance for health providers in USA to consider.

It would be good to know how these matters are addressed in other parts of the USA where there must be a number of isolated communities and people facing similar social and demographic difficulties in managing their care.

Reviewer #2: Overall this paper does a good job of illustrating unequal access to hospice and palliative care in rural areas of the SE United States.

I do have a couple of comments and observations that will make this paper stronger especially from a geographic perspective.

First the literature review section should include a review of other methods to calculate travel time - given that travel time is the deciding variable for assessing accessibility.

They could start with these papers:

1. Tansley G, Schuurman N, Bowes M, Erdogan M, Asbridge M, Yanhar N. Effect of predicted travel time to trauma care on mortality in major trauma patients in Nova Scotia. Canadian Journal of Surgery. 2018;Accepted August 28, 2018.

1. Schuurman N, Fiedler R, Grzybowski S, Grund D. Defining rational hospital catchments for non-urban areas based on travel-time. International Journal of Health Geographics. 2006;5(43).

1. Walker BB, Schuurman N, Auluck A, Lear AS, Rosin M. Socioeconomic Disparities in Head and Neck Cancer Patients’ Geographical Access to Comprehensive Cancer Treatment Centres in British Columbia, Canada. Rural and Remote Health

. 2017.

1. Tansley G, Schuurman N, Amram O, Yanchar N. Spatial access to emergency services in low- and middle-income countries: a GIS-based analysis. PLOS One. 2015.

Basically to calculate travel time, you can use origin destination matrices. Other methods such as the network analysis described here are totally acceptable but there should be some discussion of relative efficacy.

Finally the figures are (in my file anyway) totally pixelated and unreadable.

6. PLOS authors have the option to publish the peer review history of their article (what does this mean?). If published, this will include your full peer review and any attached files.

Reviewer #1: No

Reviewer #2: No

---

## [Author Response · Author response to Decision Letter 0]

13 Nov 2020

Editor’s Comments

Editor’s Comment 

Response

The manuscript has been revised to meet all of PLOS ONE’s style requirements. 

Editor’s Comment

2. Please include in your Methods section the date ranges over which you recruited participants to this study.

Response

The date ranges for the study have been added to the “Data Collection” sub-section of the “Methods” section of the manuscript.

Editor’s Comment

3. Please include additional information regarding the survey or questionnaire used in the study and ensure that you have provided sufficient details that others could replicate the analyses. If you developed and/or translated a questionnaire as part of this study and it is not under a copyright license more restrictive than Creative Commons Attribution (CC-BY), please include a copy, in both the original language and English, as Supporting Information. 

Response

- We have added additional information about the survey questionnaire including when it was done as well as information on pre-testing of the questionnaire. This information was added to the “data collection” sub-section of the “methods” section.

- We have included the questionnaire in the revised submission.

Editor’s Comment 

4.We note that you have indicated that data from this study are available upon request. PLOS only allows data to be available upon request if there are legal or ethical restrictions on sharing data publicly. For more information on unacceptable data access restrictions, please see http://journals.plos.org/plosone/s/data-availability#loc-unacceptable-data-access-restrictions.

Response De-identified study data was uploaded to Dryad as part of this resubmission and will be released upon publication of the manuscript.

The DOI for the dataset is as follows:

Hutson, Sadie (2020), End of Life Care Among Persons with HIV, Dryad, Dataset, https://doi.org/10.5061/dryad.t1g1jwt13

Comment 

Response

Figure captions of all the manuscript figures have been added to the end of the manuscript text.

Comment 

6. Please include your tables as part of your main manuscript and remove the individual files. Please note that supplementary tables (should remain/ be uploaded) as separate "supporting information" files.

Response

The table (Table 1) has been included as part of the manuscript. The original individual files has been removed from the submission.

 

Comment 

7.We note that [Figure(s) 1, 2, 3 and 4] in your submission contain [map/satellite] images which may be copyrighted. All PLOS content is published under the Creative Commons Attribution License (CC BY 4.0), which means that the manuscript, images, and Supporting Information files will be freely available online, and any third party is permitted to access, download, copy, distribute, and use these materials in any way, even commercially, with proper attribution. For these reasons, we cannot publish previously copyrighted maps or satellite images created using proprietary data, such as Google software (Google Maps, Street View, and Earth). For more information, see our copyright guidelines: http://journals.plos.org/plosone/s/licenses-and-copyright.

1. You may seek permission from the original copyright holder of Figure(s) [1, 2, 3 and 4] to publish the content specifically under the CC BY 4.0 license. 

Response 

The required copyright license has been attached as “Other” file together with the revised manuscript

---

## [Decision Letter · Decision Letter 1]

27 Nov 2020

PONE-D-20-04475R1

Geographic distribution of hospice, homecare, and nursing home facilities and access to end-of-life care among persons living with HIV/AIDS in Appalachia

PLOS ONE

Dear Dr. Hutson,

Thank you for submitting your manuscript to PLOS ONE. Your manuscript is nearly ready to accept for publication, subject to one small but important change. Please remove 'd' from the end of 'advance' in 'advance care planning' throughout. This is a common mistake, but one that PLOS ONE should not be perpetuating. 

We look forward to receiving your revised manuscript.

Kind regards,

Tim Luckett

Academic Editor

PLOS ONE

---

## [Author Response · Author response to Decision Letter 1]

27 Nov 2020

Editor’s Comment 

1. Please remove 'd' from the end of 'advance' in 'advance care planning' throughout. This is a common mistake, but one that PLOS ONE should not be perpetuating. 

Response

The manuscript has been revised to remove the ‘d’ in advanced care planning to now read, “advance care planning.” I have included a clean copy and tracked changes copy.

---

## [Editor Report · Decision Letter 2]

30 Nov 2020

Geographic distribution of hospice, homecare, and nursing home facilities and access to end-of-life care among persons living with HIV/AIDS in Appalachia

PONE-D-20-04475R2

Dear Dr. Hutson,

We’re pleased to inform you that your manuscript has been judged scientifically suitable for publication and will be formally accepted for publication once it meets all outstanding technical requirements.

Kind regards,

Tim Luckett

Academic Editor

PLOS ONE

---

## [Editor Report · Acceptance letter]

3 Dec 2020

PONE-D-20-04475R2 

Geographic distribution of hospice, homecare, and nursing home facilities and access to end-of-life care among persons living with HIV/AIDS in Appalachia 

Dear Dr. Hutson:

I'm pleased to inform you that your manuscript has been deemed suitable for publication in PLOS ONE. Congratulations! Your manuscript is now with our production department. 

Kind regards, 

on behalf of

Dr. Tim Luckett 

Academic Editor

PLOS ONE